# Iron phosphide nanocrystals as an air-stable heterogeneous catalyst for liquid-phase nitrile hydrogenation

Tomohiro Tsuda[1], Min Sheng[1], Hiroya Ishikawa[1], Seiji Yamazoe [2], Jun Yamasaki [3], Motoaki Hirayama[4,5,6], Sho Yamaguchi [1], Tomoo Mizugaki [1,7] & Takato Mitsudome [1,6] ✉

Iron-based heterogeneous catalysts are ideal metal catalysts owing to their abundance and low-toxicity. However, conventional iron nanoparticle catalysts exhibit extremely low activity in liquid-phase reactions and lack air stability. Previous attempts to encapsulate iron nanoparticles in shell materials toward air stability improvement were offset by the low activity of the iron nanoparticles. To overcome the trade-off between activity and stability in conventional iron nanoparticle catalysts, we developed air-stable iron phosphide nanocrystal catalysts. The iron phosphide nanocrystal exhibits high activity for liquid-phase nitrile hydrogenation, whereas the conventional iron nanoparticles demonstrate no activity. Furthermore, the air stability of the iron phosphide nanocrystal allows facile immobilization on appropriate supports, wherein $TiO_2$ enhances the activity. The resulting $TiO_2$-supported iron phosphide nanocrystal successfully converts various nitriles to primary amines and demonstrates high reusability. The development of air-stable and active iron phosphide nanocrystal catalysts significantly expands the application scope of iron catalysts.

Iron is the most abundant transition metal in the Earth's crust and is considered an ideal metal for catalysts owing to its extremely low cost, low-toxicity, and unique catalytic properties[1-5]. In particular, iron-based heterogeneous catalysts play a key role in two reactions vital for society, namely the Haber–Bosch process for ammonia synthesis[6-8] and the Fischer–Tropsch process for synthesizing gasoline-range iso-paraffins from syngas[9,10]. In these hydrogenation reactions, the iron oxide species deposited on metal oxide supports are reduced by $H_2$ at high temperatures to form zero-valent iron nanoparticles (Fe NPs). Thus, low-valent Fe NPs produced in situ are the true active species for the aforementioned gas-phase reactions[11,12]. However, as conventional

Fe NP catalysts require harsh reaction conditions to be active, they exhibit extremely low activity toward liquid-phase hydrogenation reactions that are conducted under mild reaction conditions (i.e., temperatures: < 473 K); such reactions are usually involved in the synthesis of bulk and fine chemicals and biomass transformation. Moreover, the Fe NPs are inherently unstable and are easily oxidized to inactive $FeO_x$ even in the presence of small amounts of oxygen[13]. Hence, the handling of such Fe NP catalysts requires strict oxygen-free conditions during all catalytic manipulation steps, including the preparation, reaction, separation, and recycling stages[14]. Furthermore, the difficulties of using these unstable Fe NPs have restricted attempts to

[1]Department of Materials Engineering Science, Graduate School of Engineering Science, Osaka University, 1-3 Machikaneyama, Toyonaka, Osaka 560-8531, Japan. [2]Department of Chemistry, Tokyo Metropolitan University, 1-1 Minami Osawa, Hachioji, Tokyo 192-0397, Japan. [3]Research Center for Ultra-High Voltage Electron Microscopy, Osaka University, 7-1 Mihogaoka, Ibaraki, Osaka 567-0047, Japan. [4]Department of Applied Physics, The University of Tokyo, 7-3-1 Hongo, Bunkyo-ku, Tokyo 113-8656, Japan. [5]RIKEN Center for Emergent Matter Science (CEMS), 2-1 Hirosawa, Wako, Saitama 351-0198, Japan. [6]PRESTO, Japan Science and Technology Agency (JST), 4-1-8 Honcho, Kawaguchi, Saitama 333-0012, Japan. [7]Innovative Catalysis Science Division, Institute for Open and Transdisciplinary Research Initiatives (ICS-OTRI), Osaka University, Suita, Osaka 565-0871, Japan. ✉e-mail: mitsudom@cheng.es.osaka-u.ac.jp

improve their catalytic performance, such as through particle size control, support alterations, and heteroatom doping[15–17]. Therefore, the preparation of Fe NP catalysts has been limited to the classical in situ reduction of Fe ions supported on metal oxides that can withstand high temperatures and $H_2$ pressures[18,19]. To address the limitation associated with the instability of Fe NPs in air, various strategies have been employed, such as coating Fe NPs with metal oxides or N-doped carbon layers[20–26]. However, this approach often results in improved air stability at the expense of decreased catalytic activity due to the shielding of active surface sites. A recent example illustrating this trade-off is the use of Fe/FeO$_x$ core–shell NPs supported on $SiO_2$ as a heterogeneous catalyst for the hydrogenation of nitriles[25]. Unfortunately, this catalyst exhibited insufficient activity and stability, representing a significant limitation of conventional iron catalysts. It is therefore vital to overcome this trade-off for the development of highly active and stable iron-based heterogeneous catalysts suitable for liquid-phase reactions.

The rapid advancement in nanotechnology over the past decade has led to the development of various synthetic techniques for nano-sized metal non-oxides, including metal nitrides, phosphides, and sulfides. These nanomaterials frequently exhibit unique catalytic properties distinct from those of conventional metal oxides and metals[27–29]. In this context, metal phosphide nanomaterials have recently emerged as highly promising catalysts, demonstrating superior activity and stability compared to conventional metal catalysts in gas-phase desulfurization[30] and liquid-phase reactions[31–33], as well as in the electrocatalytic hydrogen-evolution reaction[34]. However, the catalytic potential of iron phosphide in liquid-phase reactions remains largely unexplored, despite the high demand for iron-based heterogeneous catalysts in the field of fine and bulk chemical syntheses.

To overcome the limitation of conventional iron catalysts, our focus was on developing iron phosphide nanocrystals (Fe$_2$P NCs). The Fe$_2$P NC catalyst exhibited excellent stability and high activity for the liquid-phase hydrogenation of nitriles, a process of significant industrial importance in the production of primary amines[35–37]. The Fe$_2$P NC catalyst demonstrated high reusability without a significant loss of activity. This study represents the successful demonstration of an air-stable and reusable iron-based heterogeneous catalyst for nitrile hydrogenation.

## Results and discussion

### Synthesis and characterization of Fe$_2$P NC catalysts

The Fe$_2$P NCs were synthesized from triphenylphosphite, hexadecylamine, and Fe(CO)$_5$ in black powder form. Subsequently, the prepared Fe$_2$P NCs were immobilized on $TiO_2$, $SiO_2$, and carbon (C) supports, denoted Fe$_2$P NC/TiO$_2$, Fe$_2$P NC/SiO$_2$, and Fe$_2$P NC/C, respectively. For comparison, Fe NP/TiO$_2$ was also prepared as a model of a conventional Fe NP catalyst[38], and was used in the hydrogenation reaction without being exposed to air.

X-ray diffraction (XRD) of the prepared Fe$_2$P NCs revealed three characteristic peaks at $2\theta = 40.1$, 52.1, and 54.2°, which were assigned to the $(2\bar{1}\bar{1}1)$, (0002), and $(30\bar{3}0)$ crystalline planes of hexagonal Fe$_2$P, respectively (Supplementary Fig. 1). Transmission electron microscopy (TEM) of the Fe$_2$P NCs revealed regular nanorods with a mean size of $26.5 \times 8.7$ nm (Fig. 1a, d, and Supplementary Fig. 2). Side and top view high-resolution TEM of the Fe$_2$P NCs (Fig. 1b, e, respectively) revealed lattice spacings of 0.17 and 0.51 nm, corresponding to the (0002) and $(10\bar{1}0)$ hexagonal Fe$_2$P planes, respectively. The selected area electron diffraction (SAED) patterns from the side and top views (Fig. 1c, f, respectively) also showed the spot patterns indexed to the $[10\bar{1}0]$ and [0001] of the Fe$_2$P hexagonal crystal, proving the formation of single crystalline Fe$_2$P. Moreover, high-angle annular dark-field scanning transmission electron microscopy (HAADF-STEM) and energy-dispersive X-ray spectroscopy (EDX) of the Fe$_2$P NCs recorded from

side and top views (Fig. 1g–j and Fig. 1k–n, respectively) depicted uniform distributions of Fe and P within the Fe$_2$P NCs. EDX elemental analysis of the Fe$_2$P NCs (Supplementary Fig. 3) revealed that the molar ratio of Fe to P was close to 2:1, indicating that the nanorods formed an ideal composition. These results indicate that the prepared Fe$_2$P NCs are composed of crystalline Fe$_2$P (Fig. 1o) and have a hexagonal prism structure consisting of the $(10\bar{1}0)$ and (0001) surfaces, as represented in Fig. 1p. The TEM image and EDX maps confirmed that Fe$_2$P NCs were dispersed uniformly on the $TiO_2$ support without structural degradation (Supplementary Fig. 4).

Subsequently, X-ray photoelectron spectroscopy (XPS) was used to obtain information regarding the electronic states of the bare and supported Fe$_2$P NCs. As shown in Fig. 2a, the Fe 2$p$ spectrum of the Fe$_2$P NCs after exposure to air consisted of two peaks at 707.2 and 720.1 eV assigned to metallic Fe 2$p_{3/2}$ (706.8 eV) and Fe 2$p_{1/2}$ (720.0 eV), respectively[39]. The metallic nature of the Fe$_2$P NCs was further supported by electron energy loss spectroscopy (EELS) (Supplementary Fig. 5)[40]. These results indicate that the Fe$_2$P NCs contain air-stable low-valent Fe (i.e., Fe$^0$) species. The most intense peaks in the XPS spectra of the supported Fe$_2$P NC catalysts reflected the presence of the metallic Fe species, and the minor peaks observed for both Fe$_2$P NC/TiO$_2$ and Fe$_2$P NC/SiO$_2$ at ~710 eV corresponded to ionic Fe (i.e., FeO) species (Fig. 2b, c). The peaks attributed to FeO may represent the formation of Fe–O–Ti or Fe–O–Si bonds through the metal–support interactions (see details in Supplementary discussion). The Fe 2$p$ peaks observed for the supported Fe$_2$P NCs appear at lower energies than those of the non-supported species. Importantly, the largest shift was observed when loading the Fe$_2$P NCs onto $TiO_2$ due to electron donation from $TiO_2$ to the Fe$_2$P NCs (inset of Fig. 2e)[41]. In addition, the P 2$p$ spectrum of the Fe$_2$P NCs revealed an asymmetric peak split into two P$^0$ peaks at 129.5 eV (P 2$p_{3/2}$) and 130.3 eV (P 2$p_{1/2}$), and an additional peak at 132.1 eV corresponds to the phosphate species formed through surface oxidation (Supplementary Fig. 6)[42].

### Evaluation of catalytic property of Fe$_2$P NC in nitrile hydrogenation

The catalytic activity of the Fe$_2$P NCs was then evaluated in the hydrogenation of nitriles, which is an important reaction for the synthesis of primary amines. Although Ni- and Co-based sponge metals (Raney catalysts) are used for the hydrogenation of nitriles in industry, these catalysts are prone to significant deactivation during storage and require harsh reaction conditions due to their low activities. Thus far, various metal NP catalysts based on noble metals (i.e., Pt, Pd, Ru, Rh, Re, and Ir) and non-precious metals (i.e., Co and Ni) have been developed as alternatives to Raney catalysts[43–47]. In contrast, Fe NP catalysts are extremely rare, with only one recent report discussing the use of Fe/FeO$_x$ core–shell NPs supported on $SiO_2$ in the hydrogenation of nitriles[25]. However, the active surface Fe NPs are encapsulated with an FeO$_x$ shell, resulting in low activity, and a significant loss of its activity in the reuse experiments was caused by the low stability. Moreover, the use of Al foil or Al($i$-OPr)$_3$ as an additive is required to activate the Fe NPs. Therefore, the development of new class of Fe-based heterogeneous catalysts with high activity and stability for the nitrile hydrogenation remains in a considerable challenge.

Figure 3a shows the results of the hydrogenation of benzonitrile (1a) as a model substrate using Fe$_2$P NCs and supported Fe$_2$P NC catalysts without pretreatment under 3.8 MPa of $H_2$ and 0.2 MPa of $NH_3$ at 453 K for 2 h. Notably, Fe$_2$P NCs promoted the hydrogenation of 1a to give benzylamine (2a) in a 20% yield. In addition, the use of a $TiO_2$ support considerably improved the activity of the Fe$_2$P NCs and 2a was afforded in 78% yield, whereas the carbon, $SiO_2$, and other metal oxide supports did not lead to any significant increase in yield (Supplementary Table 1). These results suggest that the type of support affects the hydrogenation efficiency, thereby indicating the importance of the metal–support interactions. Fe$_2$P NC/TiO$_2$ also gave 2a in excellent

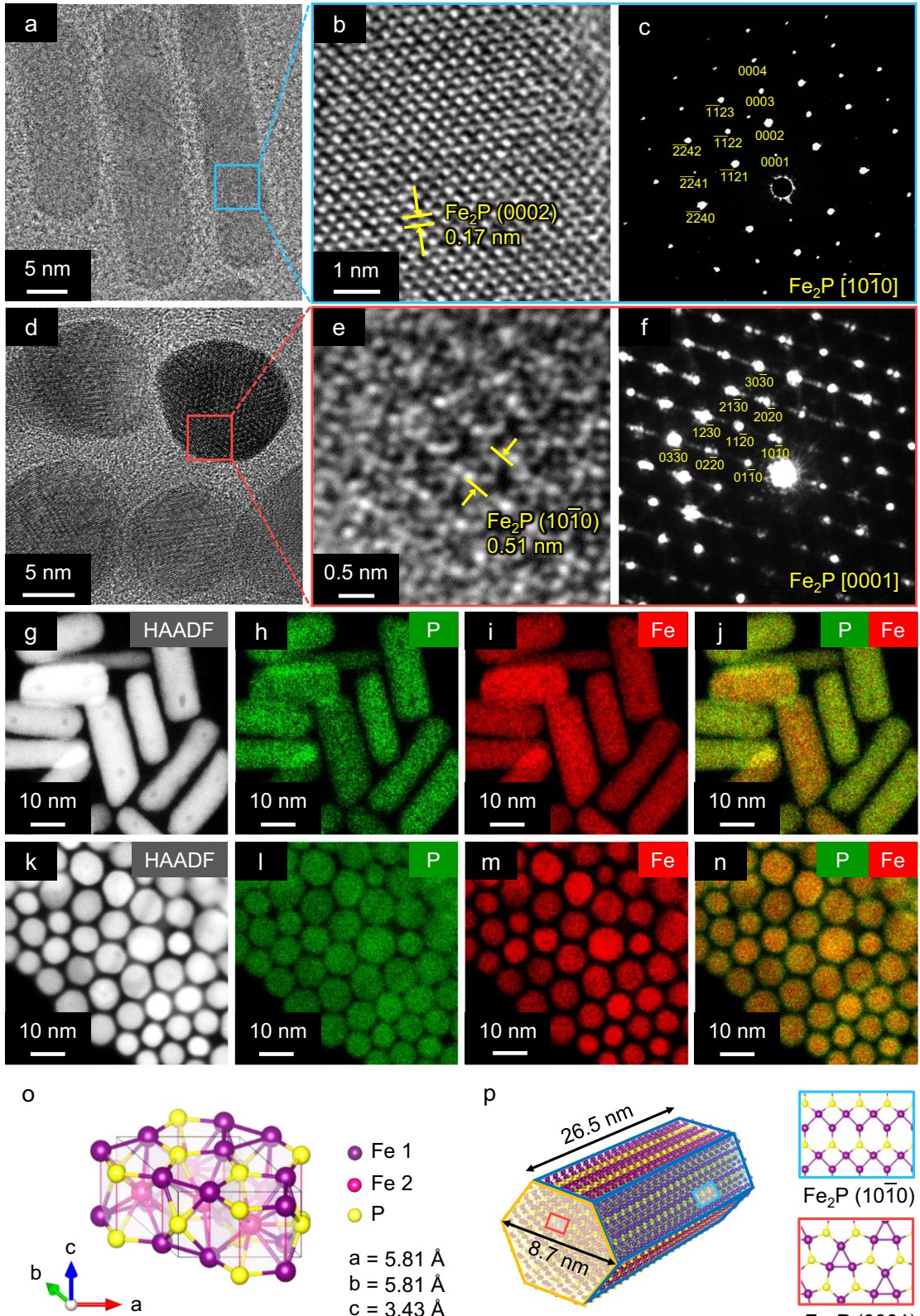

**Fig. 1 | Structural characterization of Fe₂P NCs. a**, **b** Side view TEM images of the Fe₂P NCs. **c** SAED pattern of the Fe₂P NCs indicated by the blue square in part (**a**). **d**, **e** Top view TEM images of the Fe₂P NCs. **f** SAED pattern of the Fe₂P NCs indicated by the red square in part (**d**). **g** Side view HAADF-STEM image of the Fe₂P NCs with elemental mapping images of (**h**), P and (**i**), Fe. **j** Composite overlay of parts (**h**) and (**i**). **k** Top view HAADF-STEM image of the Fe₂P NCs with elemental mapping images of (**l**), P and (**m**), Fe. **n** Composite overlay of parts (**l**) and (**m**). **o** Unit cell of Fe₂P. **p** Proposed crystal structure of the Fe₂P NCs.

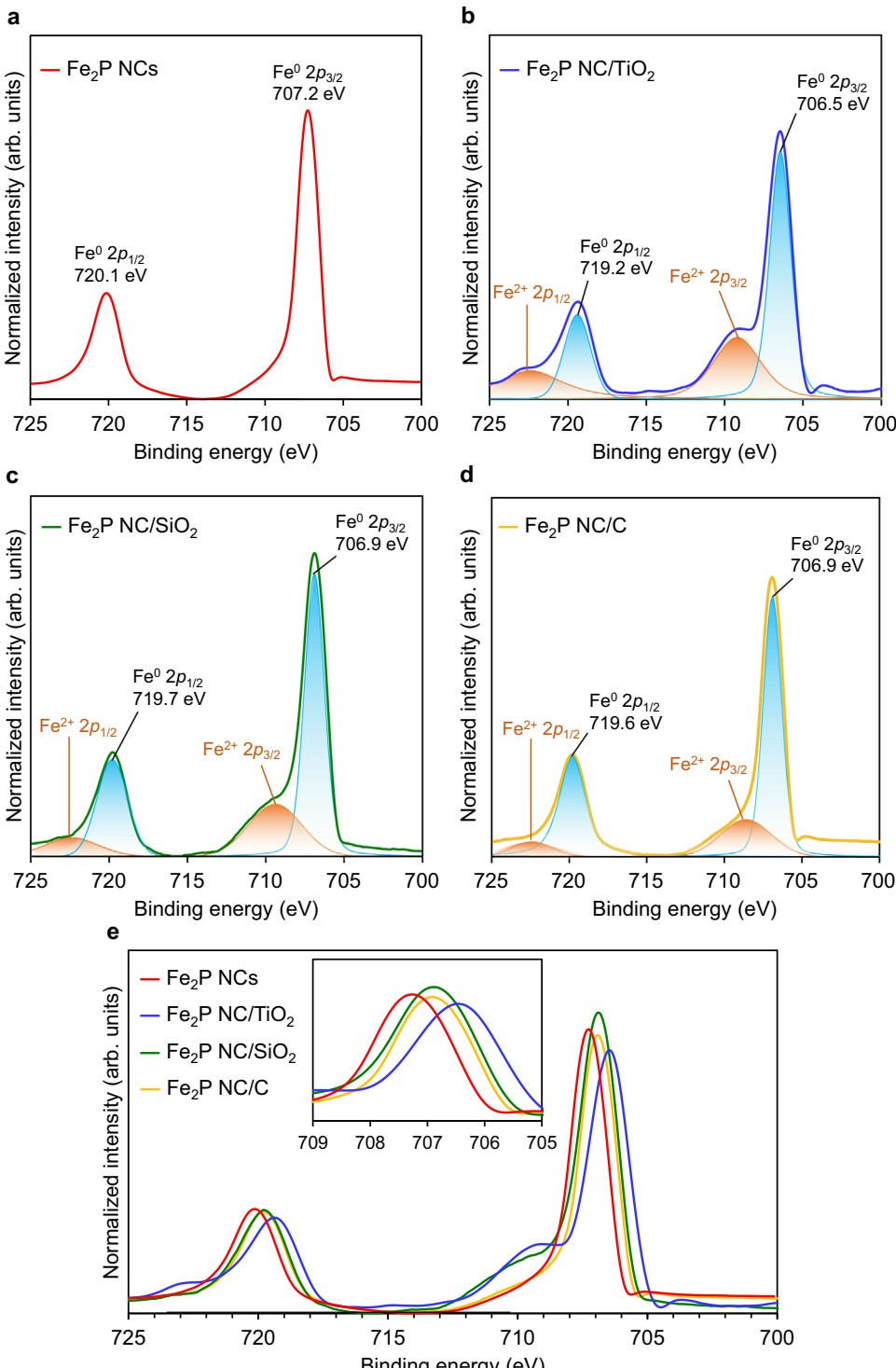

**Fig. 2 | Fe 2*p* XPS spectra of Fe₂P NCs and supported Fe₂P NC. a** Fe₂P NCs. **b** Fe₂P NC/TiO₂. **c** Fe₂P NC/SiO₂, **d** Fe₂P NC/C. **e** Composite overlay of the spectra shown in parts (**a**–**d**). The inset shows an enlarged view of the Fe 2*p*₃/₂ peaks.

yield (95%) when the reaction time was extended to 3 h. Furthermore, Fe₂P NC/TiO₂ performed well at a lower H₂ pressure of 0.5 MPa, although a longer reaction time was required (i.e., 24 h, 93% yield). In contrast, a conventionally pre-reduced Fe NP/TiO₂ catalyst and the commercially available bulk Fe₂P were inactive in this hydrogenation reaction. These results clearly demonstrate that nanosized iron phosphide species exhibit unique and excellent catalytic properties for the hydrogenation of nitriles. The effects of adding bases to the Fe₂P NC/TiO₂ reaction system were then investigated due to the fact that

bases are effective in activating nitrile hydrogenation catalysts[48,49]. As outlined in Fig. 3b, MgO was the most effective base examined, increasing the yield of **2a** from 78 to 96%, whereas other bases did not lead to any significant improvement in yield. The effect of NH₃ was also confirmed[50], wherein the addition of NH₃ enhanced the selectivity toward primary amine formation (Supplementary Figs. 7–10). In addition, we conducted the hydrogenation of **1a** using deuterium-labeled 2-propanol (2-propanol-*d₈*) as a solvent. The resulting **2a** did not show any deuterium incorporation, confirming that hydrogen

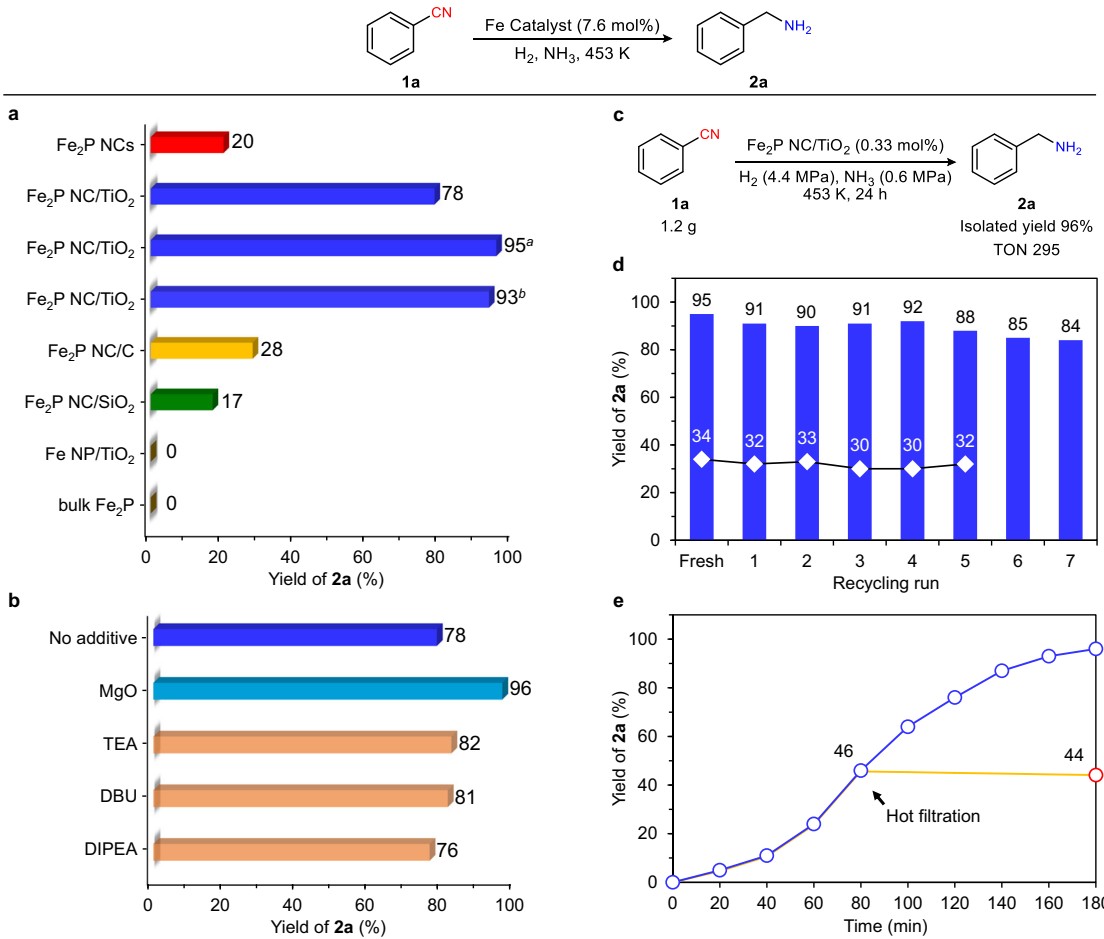

**Fig. 3 | Catalytic performance of Fe₂P NCs in the hydrogenation of 1a.**
**a** Hydrogenation of **1a** using various Fe catalysts. Reaction conditions: Fe catalyst (Fe: 7.6 mol%), **1a** (0.5 mmol), 2-propanol (3 mL), H₂ (3.8 MPa), NH₃ (0.2 MPa), 2 h. Yield was determined by gas chromatography (GC) using the internal standard technique. $^a$3 h. $^b$H₂ (0.5 MPa), 24 h. **b** Effect of bases on the hydrogenation of **1a**. Reaction conditions: Fe₂P NC/TiO₂ (0.1 g), **1a** (0.5 mmol), 2-propanol (3 mL), base (0.1 mmol), H₂ (3.8 MPa), NH₃ (0.2 MPa), 2 h. TEA: triethylamine. DBU: 1,8-diaza-bicyclo[5.4.0]undec-7-ene. DIPEA: *N,N*-diisopropylethylamine. **c** Hydrogenation

of **1a** under a high substrate/Fe ratio. Reaction conditions: Fe₂P NC/TiO₂ (0.1 g, Fe: 0.33 mol%), **1a** (11.7 mmol), 2-propanol (20 mL). **d** Reuse experiments. Reaction conditions: Fe₂P NC/TiO₂ (0.1 g), **1a** (0.5 mmol), 2-propanol (3 mL), H₂ (3.8 MPa), NH₃ (0.2 MPa). Reaction time: 3 h (blue columns), 1 h (white diamonds). **e** Hot filtration experiments. Reaction conditions: Fe₂P NC/TiO₂ (0.1 g), **1a** (0.5 mmol), 2-propanol (3 mL), H₂ (3.8 MPa), NH₃ (0.2 MPa). Blue circles: without filtration of the catalyst. Red circle: with removal of the catalyst by hot filtration after 80 min.

source is not 2-propanol but H₂ in this reaction (Supplementary Figs. 11 and 12).

Figure 3c shows the evaluation of the durability of Fe₂P NC/TiO₂ under a high substrate/Fe ratio (S/Fe = 307). 1.2 g of **1a** was converted into **2a** in 96% yield, with a turnover number (TON) of 295. This TON value is the highest reported for the iron catalysts (Supplementary Table 2). The durability of Fe₂P NC/TiO₂ was also highlighted in the recycling experiments (Fig. 3d). In contrast to the conventional air-unstable Fe catalysts with difficult handling, the spent Fe₂P NC/TiO₂ catalyst was easily recovered under ambient conditions through centrifugation and was reused without any additional treatment. Notably, Fe₂P NC/TiO₂ exhibited a consistently high activity without significant loss up to the fourth recycling experiment. The fresh and reused Fe₂P NC/TiO₂ exhibited similar reaction rates over a short reaction time (i.e., 1 h) during the recycling experiments (white diamonds in Fig. 3d), demonstrating the durability of this catalyst. The yield of **2a** slightly decreased after the fifth recycle, which may be attributed to the reduced amount of catalyst during the recovery process (Supplementary Fig. 13). In addition, the hot filtration of the Fe₂P NC/TiO₂ catalyst was carried out to separate the catalyst from the reaction mixture when the yield of **2a** reached ~50% (i.e., 80 min, Fig. 3e). The resulting filtrate was further treated under the same reaction

conditions in the absence of the filtered catalyst, and no increase in the yield of **2a** was observed. Elemental analysis of the filtrate using inductively coupled plasma-atomic emission spectrometry (ICP-AES) confirmed the absence of Fe and P species (detection limit: 0.004 ppm Fe, 0.001 ppm P). These results indicate that no leaching of the solid catalyst into the reaction solution occurred. The structure and electronic states of the used Fe₂P NC/TiO₂ were then investigated. ICP-AES revealed that the quantities of Fe and P in the Fe₂P NC/TiO₂ before and after the reaction were comparable (Supplementary Table 3), while representative TEM images showed no significant changes in the morphology of the Fe₂P NCs (Supplementary Fig. 14). In addition, the XPS spectra of Fe₂P NC/TiO₂ before and after the reaction were similar (Supplementary Fig. 15). Thus, overall, these observations confirm the excellent reusability of the Fe₂P NC/TiO₂ catalyst.

The substrate scope of the hydrogenation of various nitriles was explored using the optimized Fe₂P NC/TiO₂ catalyst and reaction conditions (Fig. 4a). Benzonitrile derivatives bearing electron-withdrawing (i.e., halogen and trifluoromethyl) or electron-donating groups (i.e., methyl, *tert*-butyl, methoxy, amino, dimethyl-lamino, methyl sulfide, phenoxy, and methylenedioxy) were converted into the corresponding benzylamines in high yields (**2a–2u**). In addition, nitriles bearing aromatic moieties, including

**Fig. 4 | Applicability of Fe$_2$P NC/TiO$_2$ in the nitrile hydrogenation. a** Substrate scope of nitriles. Reaction conditions: Fe$_2$P NC/TiO$_2$ (0.1 g), substrate (0.5 mmol), 2-propanol (3 mL), MgO (0.1 mmol), H$_2$ (3.8 MPa), NH$_3$ (0.2 MPa), 3 h. Yield was determined by GC using the internal standard technique. $^a$6 h. $^b$12 h. $^c$Fe$_2$P NC/TiO$_2$ (0.2 g), substrate (0.25 mmol), H$_2$ (4.4 MPa), NH$_3$ (0.6 MPa), 24 h. **b** Multi-gram scale nitrile hydrogenations. Reaction conditions: Fe$_2$P NC/TiO$_2$ (5 g, Fe: 1.9 mol%), substrate (5 g), 2-propanol (15 mL), H$_2$ (4.5 MPa), NH$_3$ (0.7 MPa), MgO (2.5 mmol), 453 K, 12 h. $^d$Fe: 4.3 mol%, substrate (2 g), 24 h. $^e$Fe: 7.2 mol%, substrate (1.5 g), 24 h. $^f$Fe: 4.7 mol%, substrate (2.5 g).

phenylacetonitrile, biphenylcarbonitrile, and naphthonitrile, were also successfully hydrogenated to afford the corresponding primary amines (**2v**–**2x**). Furthermore, the Fe$_2$P NC/TiO$_2$ system was suitable for use with heteroaromatic compounds, such as pyridine, indole, furan, and thiophene, without dearomatization (**2y**–**2cc**). Importantly, aliphatic and alicyclic nitriles, which are less reactive, were also hydrogenated to afford their corresponding primary amines in high yields (**2dd**–**2qq**). Notably, Fe$_2$P NC/TiO$_2$ was applicable to the hydrogenation of dinitriles, which are industrially important for the synthesis of polymer precursors. For example, isophthalonitrile, terephthalonitrile, succinonitrile, suberonitrile, sebaconitrile, and 3,3'-iminodipropionitrile were facially converted into their corresponding diamines (**2rr**–**2ww**). Additionally, the feasibility of the practical application of Fe$_2$P NC/TiO$_2$ was investigated under gram-scaled conditions using four selected nitriles, providing the corresponding primary amines in 82–98% isolated yields (Fig. 4b). These findings highlight the broad scope and practical utility of the Fe$_2$P NC/TiO$_2$ catalyst for nitrile hydrogenation.

## Discussion of the origin of catalysis by Fe$_2$P NC/TiO$_2$

As outlined in Fig. 5a, kinetic studies revealed that the initial reaction rates of the hydrogenation of **1a** using Fe$_2$P NC/TiO$_2$ increased upon raising the reaction temperature from 432 to 453 K. The corresponding Arrhenius plot showed a good degree of linearity, and the apparent

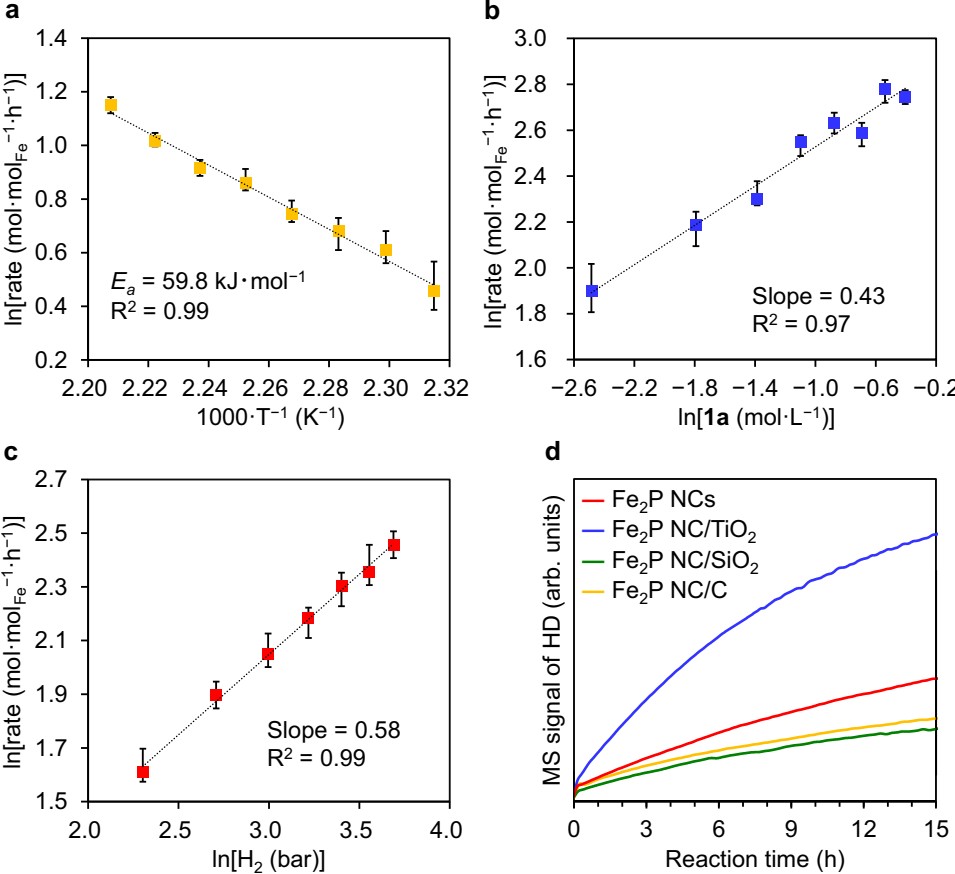

**Fig. 5 | Kinetic study of the hydrogenation of 1a and H₂–D₂ exchange experiments. a** Arrhenius plot of the hydrogenation of **1a**. Reaction conditions: Fe₂P NC/TiO₂ (0.1 g, Fe: 1.9 mol%), **1a** (2 mmol), 2-propanol (3 mL), H₂ (4 MPa), 432–453 K, 2 h. **b** Double logarithm plots of the concentration of **1a** and the initial reaction rate. **c** Double logarithm plots of the partial pressure of H₂ and the initial reaction rate.

Reaction conditions: Fe₂P NC/TiO₂ (0.1 g, Fe: 1.9–15.2 mol%), **1a** (0.25–2 mmol), 2-propanol (3 mL), H₂ (1–4 MPa), 453 K, 2 h. The plots denote the data mean values, and the error bars show the range. **d** H₂–D₂ exchange experiments. Red, blue, green, and yellow show the results obtained by Fe₂P NCs, Fe₂P NC/TiO₂, Fe₂P NC/SiO₂, and Fe₂P NC/C, respectively.

activation energy ($E_a$) was determined to be 59.8 kJ·mol⁻¹. Remarkably, this value is comparable to the activation energy of 60.3 kJ·mol⁻¹ reported for the Raney nickel catalyst[51], indicating the promising potential of Fe₂P NC/TiO₂ as a cost-effective alternative in industrial applications. Investigation of the dependency of the reaction rate on the substrate concentration and the H₂ pressure shows that the initial rates were positively correlated to both parameters (Fig. 5b, c), thereby suggesting that the rate-determining step involves the reaction of the adsorbed hydrogen species with the nitrile substrate. A hydrogen–deuterium (H₂–D₂) exchange reaction was also carried out (Fig. 5d). The Fe₂P NC catalysts promoted the H₂–D₂ exchange reaction at 453 K, confirming the H₂ activation ability of this catalyst. Notably, the H₂–D₂ exchange activity of Fe₂P NC/TiO₂ was significantly higher than that of the other Fe₂P NC catalysts, which is consistent with the superiority of Fe₂P NC/TiO₂ over the other Fe₂P NC catalysts during the nitrile hydrogenation reaction, as shown in Fig. 3a. Furthermore, as mentioned above, XPS revealed the generation of electron-rich Fe species in Fe₂P NC/TiO₂, which can be attributed to the donation of electrons from TiO₂ to the Fe₂P NCs. Such electron-rich metal species are well-known to promote the activation of H₂[52], thereby confirming the pivotal role of the TiO₂ support in improving the H₂-activation ability of the Fe₂P NCs through the formation of electron-rich Fe₂P NC; ultimately, this led to the high catalytic efficiency of this system in the nitrile hydrogenation reaction.

To gain additional insight into the origin of the hydrogenation catalysis of Fe₂P NC/TiO₂, the atomic-scale structure of the Fe species in the Fe₂P NCs was investigated using Fe *K*-edge X-ray absorption fine

structure (XAFS) analysis under an air atmosphere. Figure 6a shows the X-ray absorption near-edge structure (XANES) spectra of Fe₂P NCs and Fe₂P NC/TiO₂ along with those of the Fe foil and FeO. The absorption edge energies of the Fe₂P NCs (red line) and Fe₂P NC/TiO₂ (blue line) were considerably lower than that of FeO (orange line), and very close to that of the Fe foil (purple line), thereby suggesting that the Fe species in the Fe₂P NCs retain a metal-like state, which is consistent with the XPS result shown in Fig. 2. In addition, a Fourier transform of the extended XAFS (FT-EXAFS) spectrum of the Fe₂P NCs revealed two peaks at -1.8 and 2.3 Å, which were assigned to the Fe–P, and Fe–Fe bonds, respectively (Fig. 6b). Previous reports on metal phosphide catalysts revealed that metal–metal sites play a key role in the hydrogenation reaction[31], indicating that the Fe–Fe sites in the Fe₂P NCs can function as active sites for the nitrile hydrogenation reaction. Furthermore, the wavelet transformation (WT) results (Fig. 6c–f) showed that Fe₂P NCs and Fe₂P NC/TiO₂ produced similar patterns, which allowed the Fe–Fe and Fe–P bonds to be observed. However, their patterns were different in the regions of r = 1–1.5 Å and k = 9–12 Å⁻¹; specifically, the pattern of Fe₂P NC/TiO₂ was similar to that of FeO. This can be attributed to the Fe–O–Ti bond formation arising from an interfacial interaction within Fe₂P NC/TiO₂. This is also consistent with the XPS results shown in Fig. 2. The Fe–Fe species were further examined by EXAFS curve-fitting analysis (Supplementary Fig. 16 and Supplementary Table 4). The Fe–Fe bond length of the Fe₂P NCs was 2.65 ± 0.03 Å, which is slightly longer than that of the Fe foil (i.e., 2.48 ± 0.02 Å). Notably, the coordination number ratio of Fe–Fe to Fe–P ($CN_{Fe–Fe}/CN_{Fe–P}$) was 1.68, which is considerably smaller than the

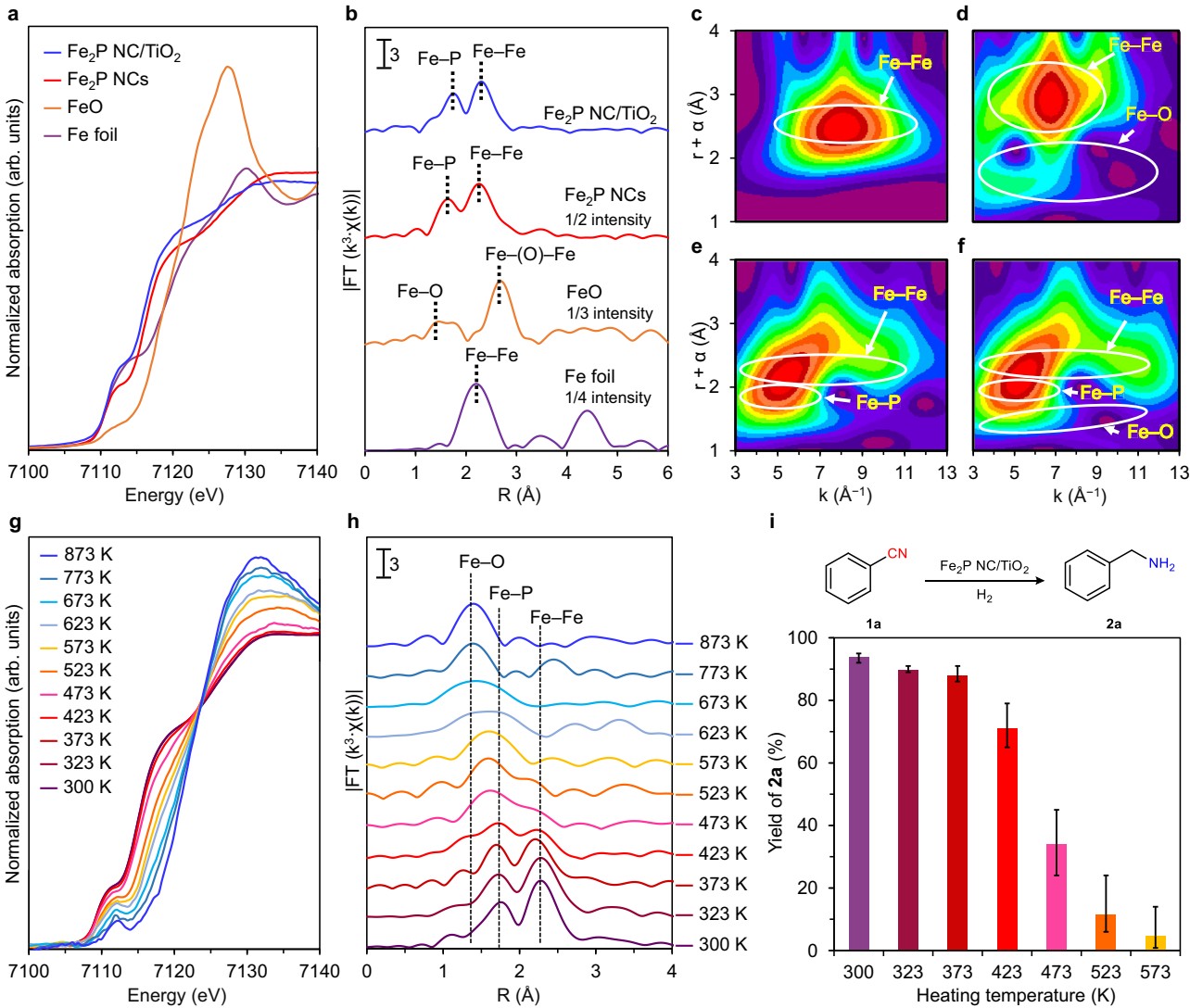

**Fig. 6 | XAFS measurements and stability evaluation of Fe₂P NC/TiO₂. a** Fe *K*-edge XANES spectra of Fe foil, FeO, Fe₂P NCs, and Fe₂P NC/TiO₂. **b** Fourier transformation of $k^3$-weighted Fe *K*-edge EXAFS spectra of Fe foil, FeO, Fe₂P NCs, and Fe₂P NC/TiO₂. Wavelet transformations for $k^3$-weighted EXAFS signals of **c** Fe foil, **d** FeO, **e** Fe₂P NCs, and **f** Fe₂P NC/TiO₂. **g** Fe *K*-edge XANES spectra and **h** Fourier transformation of $k^3$-weighted Fe *K*-edge EXAFS spectra of Fe₂P NCs with increasing

temperature under air. **i** Hydrogenation of **1a** using Fe₂P NC/TiO₂ after heat-treatment in air. The columns denote the data mean values, and the error bars show the range. Reaction conditions: Fe₂P NC/TiO₂ (0.1 g), **1a** (0.5 mmol), 2-propanol (3 mL), H₂ (3.8 MPa), NH₃ (0.2 MPa), 453 K, 3 h. Yield was determined by GC using the internal standard technique.

value in bulk Fe₂P (5.20), revealing the formation of coordinatively unsaturated Fe atoms in the Fe₂P NCs. Based on these XAFS results, coordinatively unsaturated Fe sites that can adsorb nitrile and H₂ are formed in the Fe₂P NC, which accounts for the high activity of this catalyst[53].

Finally, to investigate their structural and catalytic stability in air, the Fe₂P NCs were heated from 300 to 873 K under an air atmosphere, and their XAFS spectra and catalytic activities were evaluated at each temperature. As shown in Fig. 6g, the Fe *K*-edge XANES spectra have an isosbestic point at 7123 eV, and the absorption edge shifted to higher energies upon increasing the temperature from 423 K. The FT-EXAFS results also show a decrease in the peak intensity of the Fe–Fe bond with increasing temperature above 423 K, whereas an increase in the peak intensity of the Fe–O bond was observed due to the oxidation of the Fe₂P NCs (Fig. 6h). When these heat-treated catalysts were used in the hydrogenation of **1a**, the activity was maintained up to 423 K, and decreased gradually with further increases in temperature (Fig. 6i). These results demonstrate that both the structure and the catalytic activity of the Fe₂P NCs are highly stable in air.

In conclusion, Fe₂P NCs were synthesized and the optimal TiO₂-supported Fe₂P NCs exhibited high catalytic activity for the hydrogenation of various nitriles to provide the corresponding primary amines in high yields under liquid-phase conditions. These results contrast sharply with those of conventional iron-based heterogeneous catalysts (i.e., iron nanoparticles), which exhibited no activity in this reaction and were unstable in an air environment. Moreover, the optimal Fe₂P NC catalyst demonstrated high durability, wherein the spent Fe₂P NC catalyst was easily recovered under air, and was reusable without any significant loss of activity. Characterization using a range of spectroscopic methods revealed that coordinatively unsaturated Fe species with low oxidation states play a crucial role in the nitrile hydrogenation reaction. This study therefore demonstrates that highly active Fe₂P NCs, which do not require high-temperature pretreatment by H₂, are promising candidates to replace conventional Fe NPs. Moreover, the stability of the prepared Fe₂P NCs under air allows easy handling. In addition, further improvements to the catalytic performance can be achieved by the selection of the appropriate support material. Indeed, the use of a TiO₂ support significantly increased the hydrogenation

activity of the Fe$_2$P NC catalyst through electron donation from TiO$_2$ to Fe$_2$P NCs. While this study mainly focused on the concerted effect between Fe$_2$P NCs and TiO$_2$, it is worth noting that combining Fe$_2$P NCs with other support materials could lead to further improvements in a variety of reactions or the development of unexplored iron catalysts. Thus, the development of this catalytic system constitutes a breakthrough in terms of the stability and facile improvement of iron-based heterogeneous catalysts and significantly expands the applicability of iron catalysts beyond gas-phase hydrogenation reactions into liquid-phase organic transformations. Owing to the abundant, inexpensive, and low-toxic nature of iron, this study is expected to pave the way for establishing green, sustainable, and cost-effective methods for manufacturing valuable chemicals using iron catalysts.

## Methods

### Materials
All commercially available chemicals were used as received. Fe(CO)$_5$ (>95%) was purchased from Kanto Chemical Co., Ltd (Tokyo, Japan). Hexadecylamine (>95%) and triphenylphosphite (>97%) were acquired from Tokyo Chemical Industry Co., Ltd (Tokyo, Japan). Acetone (>99.0%), chloroform (>99.0%), ethanol (>99.5%), 2-propanol (>99.7%), Fe(NO$_3$)$_3$·9H$_2$O (>99.0%), triethylamine (>99.0%), 1,8-diazabicyclo[5.4.0]undec-7-ene (>97.0%), N,N-diisopropylethylamine (>97.0%), carbon (charcoal, activated, powder), and Fe$_2$O$_3$ were obtained from Fujifilm Wako Pure Chemical Corporation (Osaka, Japan). 2-propanol-$d_8$ (99.5%) was purchased from Sigma-Aldrich (St. Louis, the United States). TiO$_2$ (JRC-TIO-2), ZrO$_2$ (JRC-ZRO-7), and CeO$_2$ (JRC-CEO-1) were provided by the Catalysis Society of Japan (Tokyo, Japan) as reference catalysts. SiO$_2$ (CARiACT, Q-6) was supplied by Fuji Silysia Chemical Ltd (Aichi, Japan). Bulk Fe$_2$P (>99.5%, particle size: 1–5 µm) was purchased from Mitsuwa Chemicals (Osaka, Japan). MgO was obtained from Tomita Pharmaceutical Co., Ltd (Tokushima, Japan). The following nitriles were obtained commercially: Tokyo Chemical Industry Co., Ltd (Tokyo, Japan)− benzonitrile (>99%), o-tolunitrile (>98%), m-tolunitrile (>98%), p-tolunitrile (>98%), p-tert-butylbenzonitrile (>98%), o-chlorobenzonitrile (>98%), m-chlorobenzonitrile (>98%), p-chlorobenzonitrile (>98%), p-bromobenzonitrile (>98%), p-fluorobenzonitrile (>98%), 2,4-difluorobenzonitrile (>98%), 3,4-dichlorobenzonitrile (>98%), p-(trifluoromethyl)benzonitrile (>98%), p-aminobenzonitrile (>98%), p-(dimethylamino)benzonitrile (>98%), p-phenoxybenzonitrile (>98%), piperonylonitrile (>98%), o-methoxybenzonitrile (>98%), m-methoxybenzonitrile (>98%), p-methoxybenzonitrile (>98%), phenylacetonitrile (>98%), 4-cyanobiphenyl (>98%), 1-naphthonitrile (>98%), 5-cyanoindole (>98%), thiophene-2-carbonitrile (>98%), valeronitrile (>98%), octanenitrile (>97%), decanenitrile (>98%), lauronitrile (>98%), 3-ethoxypropionitrile (>99%), pivalonitrile (>98%), cyclopropanecarbonitrile (>98%), ethylene cyanohydrin (>97%), 3-methoxypropionitrile (>99%), isophthalonitrile (>98%), succinonitrile (>99%), suberonitrile (>98%), sebaconitrile (>98%), and 3,3'-iminodipropionitrile (>98%); Fujifilm Wako Pure Chemical (Osaka, Japan)−4-(methylthio)benzonitrile (>98%), 4-cyanopyridine (>98%), 3-cyanopyridine (>98%), 2-furancarbonitrile (>98%), 1-adamantanecarbonitrile (>97%), terephthalonitrile (>95%), and N-benzylidenebenzylamine (>96%); and Sigma-Aldrich (St. Louis, the United States)−cyclohexanecarbonitrile (98%). Dimethylaminopropionitrile[54], 3-[(2-hydroxyethyl)amino]propiononitrile[55], 3-isopropoxypropanenitrile[56], and benzylideneamine[57] were prepared according to the literature procedures.

The compounds employed in catalyst preparation, including iron pentacarbonyl, triphenylphosphite, and hexadecylamine were confirmed using EDX elemental analysis to have purities of >99.9%.

### General considerations
XRD analysis was carried out with Philips X'Pert-MPD (PANalytical B. V., Almelo, Netherlands) using Cu Kα radiation (45 kV, 40 mA). Elemental analysis was performed using ICP-AES (Optima 8300, Perkin Elmer, Waltham, United States) or EDX (EDX-7200, Shimadzu Corporation, Kyoto, Japan). TEM analysis was carried out with JEM-ARM200F at 200 kV (JEOL Ltd., Tokyo, Japan). STEM coupled with Super-X EDX detection with elemental mapping and EELS was carried out at 300 kV with FEI Titan Cubed G2 60-300 (FEI Co. Japan Ltd., Tokyo, Japan). Elemental EDX mapping analysis was carried out using an Esprit detector. The Fe K-edge X-ray absorption spectra were recorded at the BL01B1 and BL14B2 lines, using a Si(111) monochromator at the SPring-8 facility of the Japan Synchrotron Radiation Research Institute (Harima, Japan). The acquired EXAFS data were normalized using xTunes software[58]. The $k^3$-weighted $\chi(k)$ data of the Fe K-edge in the range of $3 \leq k \leq 13$ were Fourier transformed to the R space. WT analysis was performed in the range of $1 \leq r \leq 4$ using Morlet software to obtain information on the coordination environment of the Fe species[59]. XPS (Kratos Ultra2, Shimadzu Corporation, Kyoto, Japan) was performed using an Al Kα radiation source. The analysis area was $0.7 \times 0.3$ mm, and the C 1$s$ peak at 285.0 eV was used as the internal reference.

### Product quantification
GC–flame ionization detection (GC-2014, Shimadzu Corporation, Kyoto, Japan) was performed using an InertCap for amines (GL Sciences, Tokyo, Japan, 30 m × 0.32 mm i.d.). The oven temperature was programmed as follows: the initial temperature was 473 K and maintained for 3 min. The temperature was increased to 533 K at a rate of 20 K·min$^{-1}$, then kept constant at 533 K for 20 min. The other conditions were as follows: column flow rate (N$_2$ carrier), 2 mL·min$^{-1}$; split ratio, 17.5; vaporization chamber temperature, 523 K; and detector temperature, 533 K. The $^1$H and $^{13}$C NMR (JEOL JNM-ESC400, JEOL Ltd, Tokyo, Japan) spectra were acquired at 400 and 100 MHz, respectively. $^1$H NMR chemical shifts were reported in parts per million (ppm) using the following standard chemical shifts: tetramethylsilane (0.00 ppm), the residual proton signal in D$_2$O (4.70 ppm at 303 K), or CD$_3$OD (3.30 ppm). $^{13}$C NMR chemical shifts were reported in ppm using the following standard chemical shifts: dimethyl sulfoxide-$d_6$ (DMSO-$d_6$) (39.50 ppm), CD$_3$OD (49.00 ppm), or 1,4-dioxane-$d_8$ (67.19 ppm). NMR multiplicities were reported using the following abbreviations: s: singlet, d: doublet, dd: double doublet, t: triplet, q: quartet, sep: septet, m: multiplet, br: broad, J: coupling constants in hertz.

### Catalyst preparation

**Synthesis of the Fe$_2$P NCs.** Triphenylphosphite (10 mmol) and hexadecylamine (10 mmol) were added to a Schlenk flask and stirred at 393 K for 30 min under vacuum. After increasing the temperature to 473 K under an argon atmosphere, Fe(CO)$_5$ (1 mmol) was injected. Subsequently, the temperature was further increased to 593 K at a rate of 50 K·min$^{-1}$ and then held constant for 4 h to provide a black colloidal solution. After cooling the mixture to 298 K, the product was isolated by precipitation in acetone. Finally, the precipitate was washed with chloroform−acetone (1:1) mixture to afford the desired Fe$_2$P NCs as a black powder.

**Synthesis of the Fe$_2$P NC/support.** Fe$_2$P NCs (40 mg) were dispersed in chloroform (100 mL) and stirred with the desired support (i.e., TiO$_2$, SiO$_2$, or C, 1.0 g) at 298 K for 6 h to afford the corresponding Fe$_2$P NC/support (i.e., Fe$_2$P NC/TiO$_2$, Fe$_2$P NC/SiO$_2$, or Fe$_2$P NC/C).

**Synthesis of Fe NP/TiO$_2$.** TiO$_2$ (1.0 g) was stirred in a 2 mM ethanolic solution of Fe(NO$_3$)$_3$ (50 mL) for 48 h at 298 K, followed by evaporation at 348 K. The resulting catalyst precursor was then reduced in a H$_2$ flow with heating from 298 K to 1173 K at a rate of 5 K·min$^{-1}$, followed by holding at 1173 K for 1 h to yield the desired Fe NP/TiO$_2$.

## Typical reaction procedure

The typical reaction procedure for the hydrogenation of nitriles using $Fe_2P$ NC/$TiO_2$ was as follows. $Fe_2P$ NC/$TiO_2$ (0.1 g) was placed in a 50 mL stainless-steel autoclave with a Teflon inner cylinder, followed by the addition of nitrile (0.5 mmol) and 2-propanol (3 mL). The reaction mixture was stirred at 453 K under 3.8 MPa of $H_2$ and 0.2 MPa of $NH_3$. After 2 h of reaction, the obtained solution was analyzed by GC to determine the conversion and the yield using diethylene glycol dimethyl ether as an internal standard. In addition, to obtain the hydrochloride salt, the crude reaction mixture was filtered to remove the catalyst, and ammonia was removed under vacuum conditions. The mixture was then added to a hydrogen chloride solution (1.25 M in 1,4-dioxane), and the solvent was removed to give the pure hydrochloride salt for NMR analysis.

The yields of primary amine and imine are calculated as follows (Eqs.1 and 2):

$$\text{Yield (\%) of primary amine} = \frac{\text{mol of obtained primary amine product}}{\text{initial mol of substrate}} \times 100\% \tag{1}$$

$$\text{Yield (\%) of imine} = \frac{\text{mol of obtained imine product}}{\text{initial mol of substrate}} \times 2 \times 100\% \tag{2}$$

## Recycling experiment

After the reaction, $Fe_2P$ NC/$TiO_2$ was removed by centrifugation, and the primary amine yield was determined by GC. The spent catalyst was washed with 2-propanol for the reuse experiments. No other catalyst pretreatment was required.

## Gram-scale experiment

The gram-scale reaction of benzonitrile was performed in a 100 mL stainless-steel autoclave with a Teflon inner cylinder at 453 K according to the above procedure. After the reaction, the crude reaction mixture was filtered to remove the catalyst, and the remaining ammonia was removed under vacuum conditions. Subsequently, the mixture was added to a hydrogen chloride solution, and the solvent was removed to give the pure hydrochloride salt. The TON was calculated based on Eq.3.

$$\text{Turnover number (TON)} = \frac{\text{mol of obtained primary amine product}}{\text{mol of Fe used in the reaction}} \tag{3}$$

## Kinetic experiment

The kinetic analysis was evaluated by the conversion of benzonitrile. The initial reaction rates were determined at low conversions (<30%), and were calculated based on Eq.4.

$$\text{Reaction rate} = \frac{\text{conversion of benzonitrile} \times \text{initial mol of substrate}}{\text{mol of Fe used in the reaction} \times \text{reaction time}} \tag{4}$$

## $H_2−D_2$ exchange reaction

The $H_2−D_2$ exchange reactions were performed in a closed gas-circulation system equipped with an online quadrupole mass spectrometer (BELMass-S, BEL Japan, Inc., Osaka, Japan). The prepared $Fe_2P$ NC catalyst (Fe: 8 mg) was placed in a reactor and the air was then evacuated from the reactor under vacuum conditions. The sample was then heat-treated at 453 K under vacuum for 1 h. Subsequently, an equimolar mixture of $H_2$ and $D_2$ gases was introduced to the reaction system at 453 K and the total pressure was adjusted with Ar to 23 kPa.

The gas phase was analyzed by monitoring the signals corresponding to $m/z$ values of 2, 3, and 4.

## Data availability

The main data generated in this study are provided in the paper and the Supplementary Information. Additional data are available from the corresponding authors upon reasonable request.

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

## Acknowledgements

This work was supported by JSPS KAKENHI Grant Numbers 20H02523 (T. Mit.) and 23H01761 (T. Mit.), and JST PRESTO Grant Number JPMJPR21Q9 (T. Mit.). This study was partially supported by JST-CREST Grant Number JPMJCR21L5 (T. Mit.) and the Cooperative Research Program of the Institute for Catalysis, Hokkaido University (21A1005). The authors thank Dr. Toshiaki Ina (SPring-8) for his help with the XAFS measurements (Proposal Numbers: 2022A1117 and 2022B1585), Ryo Ota (Hokkaido University) for performing the STEM analysis, Mitsuhiko Yoshimi and Katsuyuki Hoshino (SHIMADZU CORPORATION) for performing the XPS analysis, and Prof. Akira Miura (Hokkaido University) and Prof. Kiyotaka Nakajima (Hokkaido University) for valuable discussion. The experimental analysis was supported in part by the

"Nanotechnology Platform" program at Hokkaido University (A-21-HK-0051) and the Nanotechnology Open Facilities at Osaka University (A-20-OS-0025) of the MEXT.

## Author contributions

T. Tsuda, M. Sheng, and H. Ishikawa designed the experiments, conducted the catalytic activity tests, and characterized the catalysts. S. Yamazoe performed the XAFS analysis. J. Yamasaki performed the TEM measurements and analysis. M. Hirayama, S. Yamaguchi, and T. Mizugaki discussed the experiments and results. T. Mitsudome directed the project and wrote the manuscript with input from all the authors. All authors commented critically on the manuscript and approved the final manuscript.

## Competing interests

The authors declare no competing interests.
