## [Peer Review File · Nature Communications]

Editorial Note: This manuscript has been previously reviewed at another journal that is not operating a transparent peer review scheme. This document only contains reviewer comments and rebuttal letters for versions considered at Nature Communications . Mentions of the other journal have been redacted.

REVIEWER COMMENTS

Reviewer #1 (Remarks to the Author):

I am Reviewer#4 when this manuscript was previously submitted to [Redacted]. At that time, I found this manuscript very significant and strongly recommended its publication in [Redacted]. Unfortunately, this paper was not accepted by [Redacted], but it remains a very nice work and I think it definitely deserves to be published in Nature Communication. The authors have responded well to all s even of my comments. Not only that, but I believe that the authors have conducted a lot of additional experiments and responded sincerely to the comments of the other three reviewers. As a result, I believe that the manuscript is able to more prominently showcase the strengths of this Fe₂P catalyst, which possesses both activity and stability. In response to the points raised by other reviewers, I think it is very wonderful that the authors have also carried out an industrially useful hydrogenation reactions, which is not only interesting as a basic science, but also shows its potential for industrial use of this kind of catalysts. From the above-mentioned points, I therefore recommend publishing this manuscript in Nature Communication.

Reviewer #2 (Remarks to the Author):

The authors improved their manuscript in accordance with my suggestions and Nature Commun. is well suited to publish this impressive progress.

Reviewer #3 (Remarks to the Author):

In this work, the authors have constructed an air-stable iron phosphide single nanocrystal (Fe₂P NC) catalyst for liquid-phase nitrile hydrogenation. Good air stability allows Fe₂P NC to be fixed to the TiO₂ support, and the resulting TiO₂-supported Fe₂P NC successfully converts various nitriles to primary amines and demonstrates high reusability.

The authors have carefully revised the manuscript according to reviewers' suggestions, and the manuscript could become suitable for publication in Nat. Commun. However, there are still some issues that need to be further clarified.

1. As shown in Fig. 1, Fe₂P NC is composed of [1010] and [0001] crystal planes. What is the effect of different crystal planes on the reaction performance?
2. On page 6, the data analysis of Figs 2b and 2C is inappropriate. Firstly, the synthesis of Fe₂P NC/TiO₂ is carried out at 298 K, while the strong metal–support interactions requires high temperature conditions. The synthesis temperature of Fe₂P NC/TiO₂ (298 K) is far from the requirement of strong metal–support interactions. In addition, as shown in Fig. 2d, the characteristic peak of FeO also appears in the XPS spectra of Fe₂P NC/C. I suspect that Fe₂P NC is oxidized during the synthesis of Fe₂P NC/support rather than the so-called strong metal–support interactions.
3. Please explain the oxidation resistance of Fe₂P NC/TiO₂ in air from the perspective of catalyst structure compared with other Fe based catalysts. This is the key to explain the stability of the Fe₂P NC/TiO₂.
4. The authors should describe the role of NH₃ in the reaction system in detail, rather than simply cite literature.
5. The authors think that the decrease in the activity of the cycle experiment is due to the loss of the catalyst during the recovery process. However, the results of the first five cycles are relatively stable, which seems unreasonable. Whether the decrease in catalytic activity is caused by other reasons.
6. In situ infrared can be used to detect the catalytic reaction process, and the authors can supplement the characterization to improve the reaction mechanism.
7. Since isopropyl alcohol is used as solvent in the hydrogenation of the liquid phase nitrile compound, ¹H nuclear magnetic resonance (NMR) with deuterium-labeled reagent is recommended to identify the hydrogen source.
8. In the discussion of the catalytic mechanism of Fe₂P NC/TiO₂, the authors studied the effects of different variables on the reaction, such as: hydrogen pressure, reaction temperature, and reactant concentration. The author mentioned in the article that NH₃ injection was beneficial to the occurrence of nitrile hydrogenation reaction, but did not explore the effect of the injection pressure of NH₃ on the reaction.
9. The author said in the article that “When these heat-treated catalysts were used in the hydrogenation of 1a, the activity was maintained up to 423 K” and the reaction temperature of the cycle stability test of the catalyst is 453K, which can also maintain such good durability. Are the two data inconsistent?
10. The picture in the article is not clear enough, so it is recommended to adjust it to provide readers with a better reading experience.

Reviewer #4 (Remarks to the Author):

Author have addressed all my comments with complete justifications. Now, this revised version of manuscript should be considered for the publication in Nature Communications.

Response to the reviewers' comments

Reviewer # 3

Comments:

In this work, the authors have constructed an air-stable iron phosphide single nanocrystal (Fe_2P NC) catalyst for liquid-phase nitrile hydrogenation. Good air stability allows Fe_2P NC to be fixed to the TiO_2 support, and the resulting TiO_2 -supported Fe_2P NC successfully converts various nitriles to primary amines and demonstrates high reusability. The authors have carefully revised the manuscript according to reviewers' suggestions, and the manuscript could become suitable for publication in *Nat. Commun.* However, there are still some issues that need to be further clarified.

Response: We appreciate your positive evaluation of our work and your valuable suggestions. Your expertise and thorough review have been instrumental in improving our manuscript. We have carefully considered your concerns and made the necessary revisions accordingly.

Comment 1: As shown in Fig. 1, Fe_2P NC is composed of $[10\bar{1}0]$ and $[0001]$ crystal planes. What is the effect of different crystal planes on the reaction performance?

Response 1: We appreciate this insightful comment from the reviewer regarding the effect of different crystal planes on the reaction performance of Fe_2P NC. To investigate this aspect, we conducted density functional theory (DFT) calculations. As mentioned in the original version of our manuscript, the H_2 -activation on the Fe_2P NC surface plays a crucial role in the nitrile hydrogenation (Page 16, lines 15–18 in the revised manuscript). Thus, we performed DFT calculations to examine the H_2 dissociation on Fe_2P ($10\bar{1}0$) and Fe_2P (0001) surfaces. We used Vienna Ab initio Simulation Package (VASP) as the DFT code and the PBE functional revised for solids (PBEsol) as the exchange-correlation functional, which has high reliability for structural optimization. Our calculations revealed that the binding energies for H adsorbed at Fe_2P ($10\bar{1}0$) and Fe_2P (0001) surfaces were estimated to be -20.58 and -25.31 $\text{kcal}\cdot\text{mol}^{-1}$, respectively (Fig. R1). These results indicate that H_2 activation is more favorable on the Fe_2P (0001) surface.

Fig. R1. Adsorption of hydrogen for the (10 $\bar{1}$ 0) and (0001) surfaces.

While we acknowledge the importance of understanding the effect of different crystal planes on catalytic activity, we also recognize that the data presented here represents preliminary results. To provide a more comprehensive analysis of the structure-activity relationship of Fe₂P NC, additional theoretical studies investigating factors such as Fe₂P particle size and support effects should be conducted. Therefore, we intend to conduct further investigations and present these findings in a separate paper in the near future. We believe that this explanation adequately addresses the reviewer's concerns and justifies our decision.

Comment 2: On page 6, the data analysis of Figs 2b and 2c is inappropriate. Firstly, the synthesis of Fe₂P NC/TiO₂ is carried out at 298 K, while the strong metal–support interactions require high temperature conditions. The synthesis temperature of Fe₂P NC/TiO₂ (298 K) is far from the requirement of strong metal–support interactions. In addition, as shown in Fig. 2d, the characteristic peak of FeO also appears in the XPS spectra of Fe₂P NC/C. I suspect that Fe₂P NC is oxidized during the synthesis of Fe₂P NC/support rather than the so-called strong metal–support interactions.

Response 2: Thank you for your comments and bringing up the topic of strong metal–support interaction (SMSI). We appreciate your insight and would like to clarify the usage of the term

"strong metal–support interactions" in our paper.

While we understand that SMSI is a well-established concept, first described by Tauster et al. (*J. Am. Chem. Soc.* **1978**, *100*, 170–175), as the encapsulation of metal nanoparticles by sub-oxide of supports through sintering processes, in our paper, we employed the term "strong metal–support interactions" in a broader sense to describe the interaction between the metallic Fe species and the support materials (TiO_2 and SiO_2) in the Fe_2P NC catalysts. Our intention was to convey the idea that there is a significant interaction between the metal and support, without specifically referring to the encapsulation phenomenon associated with SMSI.

We apologize if the use of the term caused any confusion. Considering your valuable suggestion, we will use "metal–support interactions" instead of "strong metal–support interactions" to avoid any misconceptions among the readers. We appreciate your guidance in enhancing the clarity of our work.

Comment 3: Please explain the oxidation resistance of Fe_2P NC/ TiO_2 in air from the perspective of catalyst structure compared with other Fe based catalysts. This is the key to explain the stability of the Fe_2P NC/ TiO_2 .

Response 3: We thank the reviewer for the insightful comment regarding the oxidation resistance of Fe_2P NC/ TiO_2 in air. In response to this comment, we conducted first-principles calculations using VASP, as mentioned in Comment 1. The DFT calculations provided valuable insights into the charge density distribution of Fe_2P . We observed a strong covalent bond network between the Fe and P atoms, as shown by the dark blue color in Fig. R2. This robust covalent bond network in Fe_2P distinguishes it from pure Fe. The presence of this strong covalent bond network in Fe_2P may impart excellent oxidation resistance to the material, making it less susceptible to oxidation compared to Fe alone. This unique structural feature is crucial in understanding the enhanced stability of the Fe_2P NC/ TiO_2 catalyst in air.

Fig. R2. Charge density distribution of Fe_2P .

This discovery of the strong covalent bond network in Fe_2P is intriguing and warrants further investigations. To gain a more comprehensive understanding of the stability of the Fe_2P NC/ TiO_2

catalyst, additional experimental and theoretical studies incorporating a broader range of factors and parameters are needed. Due to the primitive nature of the current theoretical data and the need for comprehensive results, we have decided not to include these findings in the revised manuscript. Instead, we intend to pursue more in-depth research and publish a separate paper dedicated to addressing the complex structure-activity relationships of Fe₂P NC catalysts. This response addresses the reviewer's query and clarifies our decision. Thank you for your valuable feedback, which has strengthened the quality of our study.

Comment 4: The authors should describe the role of NH₃ in the reaction system in detail, rather than simply cite literature.

Response 4: Thank you for this comment. We investigated the influence of NH₃ on the selectivity of benzylamine (**2a**) in the hydrogenation of benzonitrile (**1a**) as shown in Supplementary Fig. 7 (Fig. R3) in the original version of manuscript. In the absence of NH₃, the selectivity of **2a** considerably decreased. This decrease was attributed to the formation of the *N*-benzylidenebenzylamine (**1c**), which occurs when **2a** attacks the primary imine intermediate, resulting in the concurrent release of NH₃. To address this issue, we introduced NH₃ into the reaction system. The addition of NH₃ effectively suppressed the deammoniation process, thereby enhancing the selectivity of **2a**.

Fig. R3 (Supplementary Fig. 7). Time course of the hydrogenation of **1a** using Fe₂P NC/TiO₂ (a) in the presence or (b) absence of NH₃. Reaction conditions: Fe₂P NC/TiO₂ (0.1 g, Fe: 7.6 mol%), **1a** (0.5 mmol), 2-propanol (3 mL), H₂ (3.8 MPa), 453 K. Conversion and yield were determined by gas chromatography using an internal standard technique.

Comment 5: The authors think that the decrease in the activity of the cycle experiment is due to the loss of the catalyst during the recovery process. However, the results of the first five cycles are

relatively stable, which seems unreasonable. Whether the decrease in catalytic activity is caused by other reasons.

Response 5: We appreciate this insightful comment from the reviewer. In response, we conducted the additional cycle experiment with meticulous attention to the potential loss of catalyst during the recovery.

Figure R4 presents the results of these experiments, showing that the first four cycles exhibit relative stability, with gradual and slight decreases in the yield of **2a** after the fifth cycle. This trend closely aligns with the findings reported in the original manuscript, although the decrease has been mitigated by the significant care taken to minimize catalyst loss during recovery.

To investigate the underlying cause of this slight decrease in the yield of **2a**, we calculated the turnover number (TON) based on the amount of iron in the recovered $\text{Fe}_2\text{P}/\text{TiO}_2$ catalyst. Despite the observed decrease in yield of **2a** (indicated by blue bars), the TON values for each cycle remained relatively consistent (represented by green squares). These results provide further evidence that the decrease in yield primarily arises from the minor loss of catalyst during the recovery process. We have included these findings in the revised Supplementary Information.

Once again, we thank the reviewer for raising this point and allowing us to enhance the clarity of our study.

Fig. R4. Reuse experiments. Reaction conditions: $\text{Fe}_2\text{P NC}/\text{TiO}_2$ (0.1 g), **1a** (0.5 mmol), 2-propanol (3 mL), H_2 (3.8 MPa), NH_3 (0.2 MPa), 453 K, 3 h.

Comment 6: In situ infrared can be used to detect the catalytic reaction process, and the authors can supplement the characterization to improve the reaction mechanism.

Response 6: We appreciate your valuable comment. We conducted in-situ IR analysis of the hydrogenation reaction of benzonitrile using $\text{Fe}_2\text{P NC}$. However, we encountered limitations in

sensitivity due to the low adsorption ability and black color of the Fe₂P catalyst. Consequently, we were unable to obtain clear spectral data for benzonitrile adsorbed on Fe₂P NC. Although in-situ IR analysis would have been valuable for directly monitoring the catalytic reaction process, the properties of the Fe₂P catalyst hindered its applicability.

To compensate for this limitation, we performed control experiments to supplement the characterization and enhance our understanding of the reaction mechanism. We propose the reaction pathway depicted in Scheme R1 (Monguchi, Y. *Science* **2022**, 376, 1382–1383). In order to elucidate the details of the reaction mechanism, we conducted the hydrogenation of expected imine intermediates (benzylideneamine [**1b**] and *N*-benzylidenebenzylamine [**1c**]) as starting materials.

Scheme R1. A plausible reaction pathway for the hydrogenation of **1a** to **2a**.

Schemes R2a and b demonstrate the results of the hydrogenation of **1b** and **1c** as substrates using Fe₂P NC/TiO₂. Fe₂P NC/TiO₂ facilitated the hydrogenation of **1b** to give desired product **2a** in 89% along with the production of *N*-benzylidenebenzylamine (**1c**). Furthermore, **1c** was efficiently transformed to **2a** with high yield (Scheme R2b). These results suggest a plausible reaction pathway: the hydrogenation of **1a** produces **1b**, which is then hydrogenated to **2a**. Subsequently, **1c** is generated through the condensation of **1b** and **2a**, followed by its decomposition to produce **2a**.

Scheme R2. a, Hydrogenation of **1b**. Reaction conditions: Fe₂P NC/TiO₂ (0.1 g, Fe: 7.6 mol%), **1b** (0.5 mmol), 2-propanol (3 mL), H₂ (3.8 MPa), NH₃ (0.2 MPa), 453 K, 3 h. **b**, Hydrogenation of **1c**. Reaction conditions: Fe₂P NC/TiO₂ (0.1 g, Fe: 7.6 mol%), **1c** (0.5 mmol), 2-propanol (3

mL), H₂ (3.8 MPa), NH₃ (0.2 MPa), 453 K, 3 h.

We have included these additional results in the revised Supplementary Information. Although in-situ IR analysis was not feasible in our catalyst system, these control experiments provide valuable insights into the reaction pathway and enhance our understanding of the catalytic process. Thank you once again for your insightful comment.

Comment 7: Since isopropyl alcohol is used as solvent in the hydrogenation of the liquid phase nitrile compound, ¹H nuclear magnetic resonance (NMR) with deuterium-labeled reagent is recommended to identify the hydrogen source.

Response 7: We thanks for your insightful comment. We appreciate your suggestion to investigate the possibility of isopropyl alcohol (2-propanol) serving as a hydrogen source in the nitrile hydrogenation.

According to the reviewer's suggestion, we have conducted the hydrogenation of benzonitrile using deuterium-labeled 2-propanol (2-propanol-*d*₈) as a solvent. After the reaction, we analyzed the hydrochloride salt of the product using ¹H NMR spectroscopy. As depicted in Fig. R5, it was observed that the resulting amine product did not show any deuterium incorporation.

Fig. R5. ^1H NMR spectrum of **2a**-hydrochloride synthesized from the hydrogenation of **1a** using (a) 2-propanol or (b) 2-propanol- d_8 as solvent.

To further examine whether 2-propanol acted as the reductant, we performed the hydrogenation of benzonitrile in 2-propanol under argon atmosphere. As shown in Scheme R3, $\text{Fe}_2\text{P NC/TiO}_2$ did not promote the hydrogenation of benzonitrile under argon atmosphere. These results clearly show that 2-propanol does not act as a hydrogen source in this reaction.

Scheme R3. Hydrogenation of **1a** to **2a** under Ar atmosphere. Reaction condition: $\text{Fe}_2\text{P NC/TiO}_2$ (0.1 g, Fe: 7.6 mol%), **1a** (0.5 mmol), 2-propanol (3 mL), Ar (1.0 MPa), 453 K, 3 h.

We have included these findings in the revised version of the manuscript, highlighting that 2-propanol does not function as a hydrogen source. Thank you for bringing this aspect to our attention.

Comment 8: In the discussion of the catalytic mechanism of $\text{Fe}_2\text{P NC/TiO}_2$, the authors studied the effects of different variables on the reaction, such as: hydrogen pressure, reaction temperature, and reactant concentration. The author mentioned in the article that NH_3 injection was beneficial to the occurrence of nitrile hydrogenation reaction, but did not explore the effect of the injection pressure of NH_3 on the reaction.

Response 8: Thank you for your comment. According to the suggestion by the reviewer, we conducted additional experiments to investigate the influence of NH_3 injection pressure on the yield of the primary amine product in the hydrogenation of benzonitrile. The result shown in Fig. R6 demonstrated that the increase of NH_3 pressure enhanced the yield of the primary amine product. However, beyond an NH_3 injection pressure of 0.1 MPa, there was no further improvement in the amine yield. These results indicate that 0.1 MPa of NH_3 pressure is sufficient to promote the desired reaction.

We have included this data in the revised Supplementary Information. Thank you for bringing this aspect to our attention.

Fig. R6. Dependency of NH₃ partial pressure in hydrogenation of **1a**. Reaction conditions: Fe₂P NC/TiO₂ (0.1 g, Fe: 7.6 mol%), **1a** (0.5 mmol), 2-propanol (3 mL), H₂ (3.8 MPa), 453 K, 2 h. The columns denote the data mean values, and the error bars show the range.

Comment 9: The author said in the article that "When these heat-treated catalysts were used in the hydrogenation of **1a**, the activity was maintained up to 423 K" and the reaction temperature of the cycle stability test of the catalyst is 453 K, which can also maintain such good durability. Are the two data inconsistent?

Response 9: Thank you for your comment. We appreciate the opportunity to address this potential misunderstanding.

To clarify, the heat-treatment mentioned in the article refers to the process of treating the catalyst under air atmosphere to assess its air stability. After this heat-treatment, the Fe₂P NC catalyst was utilized in the hydrogenation of benzonitrile as described in Fig. 6i in the original manuscript. We found that even after the catalyst underwent heat-treatment up to 423 K under air atmosphere, it maintained high catalytic activity in the nitrile hydrogenation.

On the other hand, the cycle stability test was conducted at 453 K using the Fe₂P NC catalyst without any heat-treatment before use. This test was designed to evaluate the durability of the catalyst during repeated cycles of the nitrile hydrogenation. Therefore, there is no inconsistency between the two sets of data. The heat-treatment in air was performed to assess air stability, while the cycle stability test was conducted without any heat-treatment before use. These experiments provide different aspects of the catalyst's performance.

Taking into account these details, we assure you that our experimental results are consistent

and accurately reflect our findings. Thank you for bringing up this point, allowing us to clarify any potential confusion.

Comment 10: The picture in the article is not clear enough, so it is recommended to adjust it to provide readers with a better reading experience.

Response 10: We appreciate the reviewer's suggestion to improve the clarity of the graphics in our paper. We have thoroughly reviewed the manuscript and made the necessary revisions. We have improved the clarity of Figs. 1a-1n, Figs. 2, 3, Supplementary Figs. 2, 4, 6, 7 and 11. We have also made adjustments to the text color to ensure better readability for the readers.

We believe that these revisions will enhance the visual presentation of our paper and facilitate a better understanding of our research findings. Thank you for your suggestion in helping us enhance the quality of our publication.

REVIEWERS' COMMENTS

Reviewer #3 (Remarks to the Author):

The author has responded reasonably to the comments raised and made a series of modifications to the content of the manuscript. I think the manuscript is well organized and can be published.

It should be noted that in Comment 2, a characteristic peak of FeO appeared in the XPS spectrum of Fe₂P NC/C, which is suspected to be oxidized during the synthesis process of Fe₂P NC/support. Although the author replaced with "metal support interaction", it seems that the author did not provide a reasonable answer to this question.

Response to the reviewers' comments

Reviewer # 3

Comments: The author has responded reasonably to the comments raised and made a series of modifications to the content of the manuscript. I think the manuscript is well organized and can be published.

It should be noted that in Comment 2, a characteristic peak of FeO appeared in the XPS spectrum of Fe₂P NC/C, which is suspected to be oxidized during the synthesis process of Fe₂P NC/support. Although the author replaced with "metal support interaction", it seems that the author did not provide a reasonable answer to this question.

Response: We extend our sincere appreciation for the thoughtful assessment and constructive recommendations provided by Reviewer #3. We acknowledge the positive evaluation of the progress made in our manuscript. We understand the reservations expressed in Comment 2 regarding the emergence of the distinct FeO peak in the XPS spectrum of Fe₂P NC/C.

We fully recognize the difficulty associated with accurately distinguishing the origin of a peak, whether it arises from surface oxidation or metal-support interaction facilitated through oxygen species of supports.

As illustrated in Fig. 2 of the manuscript, our XPS analyses on Fe₂P NC/C, Fe₂P NC/TiO₂, and Fe₂P NC/SiO₂ revealed that the most prominent peaks represent metallic Fe species. On the other hand, the less pronounced peaks correspond to ionic Fe (FeO) species. Notably, the peak intensities of the ionic Fe species in Fe₂P NC/TiO₂ and Fe₂P NC/SiO₂ are considerably higher than those in Fe₂P NC/C. This difference could be due to the oxygen-rich nature of TiO₂ and SiO₂ surfaces in comparison to carbon, potentially suggesting the formation of Fe–O–X bonds (with X being Ti, Si, or C), which are facilitated by metal-support interactions. The relatively low peak intensity corresponding to Fe–O–C bonds may be attributed to the interaction between Fe and residual oxygen sites present on the partially oxidized carbon support. This interpretation is supported by existing literatures, which acknowledge that low valent Fe species are loaded on carbon support through Fe–O–C bonding.^{1,2}

Furthermore, we also anticipated the negative impact of surface oxidation on the catalytic performance of Fe₂P NC. However, as depicted in Fig. 3 of our manuscript, the catalytic activity of Fe₂P NC/C is higher than that of unsupported Fe₂P NC (Fe₂P NC/C and Fe₂P NC afforded 28% and 20% product yields, respectively, in the hydrogenation of benzonitrile). This experimental result supports our interpretation that the peak attributed to ionic Fe species is more indicative of

the formation of Fe–O–C bonds, rather than surface oxidation.

Reference

[1] I. Hussain et al. *Chem. Eng. J.* **2017**, *311*, 163–172.

[2] R. Zhang et al. *Sustainability* **2022**, *14*, 9324.

We acknowledge the importance of the reviewer's suggestion and the above discussion. Therefore, in response to the significance of reviewer's insight, we have thoughtfully integrated this discussion into the revised Supplementary Information, as presented below. We firmly believe that this explanation adequately addresses the reviewer's concerns and substantiates our stance.

“In relation to the XPS analysis, as pointed out by one of the reviewers, the appearance of a FeO peak in the XPS spectrum of Fe₂P NC/C has raised suspicions of oxidation during the synthesis process of Fe₂P NC/support, as opposed to metal-support interaction.

We acknowledge these concerns. Generally, distinguishing whether a characteristic peak of a metal ion originates from surface oxidation or metal-support interaction through oxygen species of supports poses significant difficulty.

As depicted in Fig. 2 of the manuscript, the XPS spectra of Fe₂P NC/C, Fe₂P NC/TiO₂, and Fe₂P NC/SiO₂ show minor peaks associated with ionic Fe species (FeO). Among them, the intensity of ionic Fe species in Fe₂P NC/TiO₂ and Fe₂P NC/SiO₂ surpasses that of Fe₂P NC/C, probably due to the oxygen-rich nature of TiO₂ and SiO₂ surfaces in comparison to carbon. This discrepancy suggests the formation of Fe–O–X bonds (where X = Ti, Si, or C) which are facilitated by metal-support interactions. The relatively lower intensity of the peak corresponding to Fe–O–C bonds might be attributed to interactions between Fe and residual oxygen sites on the partially oxidized carbon support.

Furthermore, it's important to consider the expected negative effect of surface oxidation on the catalytic performance of Fe₂P NC. Unexpectedly, Fig. 3 demonstrates that the catalytic activity of Fe₂P NC/C exceeds that of unsupported Fe₂P NC (with product yields of 28% and 20%, respectively, in the hydrogenation of benzonitrile). This experimental result provides additional support for the interpretation that the peak attributed to ionic Fe species more likely signifies the formation of Fe–O–C bonds, rather than surface oxidation. It is acknowledged that low valent Fe species are loaded on carbon support through Fe–O–C bonding^{S35,S36}.

Based on these collective findings, we currently conclude that the minor peaks corresponding to ionic Fe species likely arise from the formation of Fe–O–X bonds (where X = Ti, Si, or C) facilitated by metal-support interactions."

References

- [S35] Hussain, I. et al. Insights into the mechanism of persulfate activation with nZVI/BC nanocomposite for the degradation of nonylphenol. *Chem. Eng. J.* 311, 163–172 (2017). [10.1016/j.cej.2016.11.085](https://doi.org/10.1016/j.cej.2016.11.085).
- [S36] Zhang, R. et al. Remediation and optimisation of petroleum hydrocarbon degradation in contaminated water by persulfate activated with bagasse biochar-supported nanoscale zerovalent iron. *Sustainability* 14, 9324 (2022). [10.3390/su14159324](https://doi.org/10.3390/su14159324).